# A Large Ovarian Endometrioma Occupying the Abdominal Cavity in a Postmenopausal Patient: A Case Report

**DOI:** 10.3390/medicina59081398

**Published:** 2023-07-30

**Authors:** Maria Themeli Zografou, Antoine Naem, Antonio Simone Laganà, Harald Krentel

**Affiliations:** 1Department of Obstetrics, Gynecology, Gynecologic Oncology and Senology, Bethesda Hospital Duisburg, 47053 Duisburg, Germany; maria.zografou@yahoo.com (M.T.Z.); krentel@cegpa.org (H.K.); 2Faculty of Mathematics and Computer Science, University of Bremen, 28359 Bremen, Germany; 3Department of Health Promotion, Mother and Child Care, Internal Medicine and Medical Specialties (PROMISE), University of Palermo, 90133 Palermo, Italy; antoniosimone.lagana@unipa.it

**Keywords:** endometriosis, endometrioma, menopause, postmenopause, ovarian cyst

## Abstract

Endometriosis is defined by the presence of endometrial-like glands and/or stroma outside the uterus. The prevalence of endometriosis in postmenopausal women is reported to be 2.55%, which is much lower than that in reproductive-aged women. Ovarian endometriomas are the most common form of endometriosis. However, these form only 4.3% of ovarian masses in patients in the sixth decade of life. In this manuscript, we report the case of a 60-year-old patient who was referred to our department with an external diagnosis of an abdominal mass. The patient was in good general condition and asymptomatic. A computed tomography scan revealed the presence of a cystic mass originating from the right adnexa and measuring 26 cm. No signs of malignancy were observed. Due to the cyst’s size, a midline laparotomy and a bilateral salpingo-oophorectomy were performed successfully. A postoperative histopathologic examination confirmed the diagnosis of an ovarian endometrioma with no signs of hyperplasia or atypia. Cases of postmenopausal large ovarian endometriomas are few. However, due to the risk of malignant transformation, an oophorectomy could be considered the treatment of choice, even in asymptomatic patients.

## 1. Introduction

Endometriosis is defined by the presence of endometrial-like glands and stroma outside of the uterus [1]. Stromal endometriosis, on the other hand, is a less common manifestation of endometriosis, in which the endometriotic stroma exists without the glandular component [2]. Endometriosis is known to be an estrogen-dependent chronic inflammatory disease owing to the fact that estrogen triggers lesional growth and progression and contributes to macrophage recruitment [3]. This notion is further supported by the observation of a peak in the prevalence of endometriosis in women of childbearing age [4]. Endometriosis is reported to affect around 10% of reproductive-aged women worldwide [5] compared to 2.55% of postmenopausal women [6]. This decline is thought to be due to the hypoestrogenic environment experienced in menopause. Nevertheless, endometriosis is capable of producing estrogen locally because of its intrinsic endocrine activity [7]. Aromatase is heavily expressed in endometriosis and converts cholesterol to estrogen under the continuous stimulation of prostaglandin E2 (PGE2) [7,8]. Therefore, it is unclear whether the extrinsic hypoestrogenic milieu seen in menopause is the actual explanation of the fewer postmenopausal cases, or this could be attributed to other unknown reasons.

Ovarian endometriomas are the most common form of endometriosis [9]. These are reported to affect up to 44% of endometriosis patients [10]. In the fourth decade of life, ovarian endometriomas form around 27% of ovarian masses, and only for 4.3% in the sixth decade of life [11]. Although rare, some ovarian endometriomas can reach large sizes and extend to the upper abdomen [12]. Moreover, individual reports documented cases of large postmenopausal endometriomas that had non-ovarian origins [13]. Despite the dilemma of whether postmenopausal endometriosis is pre-existent endometriosis or de novo lesions [14], it does not seem that large lesions with benign histology are newly formed. In this paper, we report a case of a large ovarian postmenopausal endometrioma and discuss the possible pathophysiologic mechanisms of endometriosis progression and appropriate therapy in such cases.

## 2. Case Presentation

A 60-year-old patient was referred to our department with the diagnosis of a large intra-abdominal mass. The patient was in good general status and did not report symptoms aside from abdominal distention. A transabdominal ultrasonographic scan demonstrated a unilocular cyst with the absence of a suspicious Color Doppler flow. A computed tomography (CT) scan revealed a unilocular cystic mass that originated from the region of the right adnexa and occupied almost the entire abdomino-pelvic cavity. The cyst measured 26 cm at its largest dimension (Figure 1). Ascites was not observed radiologically. The CT scan combined with the ultrasonography findings decreased the suspicion of malignancy.

The laboratory results were within the normal ranges. However, the serum levels of CA-125 and CEA were 512.9 U/mL and 6.7 U/mL, respectively. The serum level of CA 15-3 was within the normal limits (17.3 U/mL). Table 1 demonstrates the complete preoperative laboratory assessment of the patient. On that basis, a borderline ovarian tumor could not be ruled out with high confidence. After appropriate patient counseling, the decision of surgical removal of both adnexae through a midline laparotomy was made. The intra-operative findings were in line with the preoperative imaging results. A large cyst originating from the right ovary was detected. The left adnexa, uterus, and peritoneum were unremarkable. A right adnexectomy was performed, and the specimen was immediately sent for a frozen section examination. The frozen section biopsy results ruled out the diagnosis of an ovarian borderline tumor and suggested ovarian endometrioma as a differential diagnosis with no signs of malignancy. Therefore, a bilateral salpingo-oophorectomy was carried out successfully. The postoperative recovery was uneventful. Upon gross examination, the cyst weighed 7630 g and had smooth surfaces without nodules or projections. A microscopic examination of the cyst’s wall confirmed the diagnosis of an ovarian endometrioma with no signs of hyperplasia or atypia (Figure 2).

## 3. Discussion

In this report, we presented a postmenopausal right ovarian endometrioma that reached a large size in an asymptomatic patient. The cyst was diagnosed with a CT scan and an ultrasound and treated surgically through a bilateral salpingo-oophorectomy. The cyst measured 26 cm and weighed around 7.5 kg. To the best of our knowledge, only few cases of similar clinical course have been reported, making this manifestation of postmenopausal endometriosis very rare. One report documented an ovarian endometrioma of 65 cm that weighed 214 kg [12]. This case is thought to be the largest ovarian endometrioma documented in the English literature [12,15]. Another report by Raju et al. [16] documented a 30 cm endometrioma in a 53-year-old patient that contained 5 L of clear, serous fluid. Yahya et al. [15] reported a case of a 30 cm ovarian endometrioma that contained 5000 mL of dark brown, cystic fluid in a 33-year-old patient. Moreover, endometriomas can be located in the abdominal cavity with remarkable adherence to the omentum but without any connection to the adnexa. In the report of Naem et al. [13], a free abdominal endometrioma reached 45 cm at its largest diameter and weighed 4.5 kg in a 67-year-old patient. All of these cases were treated surgically and had no signs of malignant transformation of the endometriosis. The risk of malignant transformation in ovarian endometriosis is reported to be 0.7–1% [17,18]. It is noteworthy that benign ovarian endometriomas have been found to harbor some cancer-associated mutations in their glandular component. These mutations are similar to those found in endometrial cancer but with lower frequency [19,20]. Such genetic alterations (especially cancer-associated mutations, like PTEN, ARID1A1, PIK3CA, and K-RAS) are thought to control the growth, invasion, symptomatology, and probably the malignant transformation of endometriotic lesions [21]. It would be interesting to study the genomes of such large cysts to see whether they harbor higher or different types of mutations that make them reach such large sizes. It is believed that the worsening of pre-existing dysmenorrhea or dyspareunia could be related to the malignant transformation of endometriosis [22]. It is noteworthy that magnetic resonance imaging is a reliable tool to diagnose endometriosis-associated cancer when suspected [23]. This can be done through the visualization of solid components, papillary projections, and mural nodules in a cyst’s wall [23].

The exact preventive measures for the progression of postmenopausal endometriomas are not known precisely yet. In addition, efficient prevention relies on a good understanding of the mechanisms leading to the progression of endometriosis. Therefore, the only recommendation to prevent such cases of large endometriomas is adhering to yearly follow-up visits to the gynecologist, with an early intervention whenever continuous or accelerating growth of ovarian endometriomas are observed. It would be useful to measure the serum levels of CA-125, CEA and CA 15-3 to investigate the chances of malignancies when lesions reach large sizes.

Although rare, surgical treatment of postmenopausal ovarian endometriomas is considered the first-line therapeutic approach to avoid the risk of malignant transformation [14], especially when a lesion is larger than 7 cm [24]. In some patients who refuse surgery or those who cannot undergo surgery, medical treatment with aromatase inhibitors could be a safe alternative to surgery [25].

## 4. Conclusions

Large ovarian endometriomas are a very rare form of postmenopausal endometriosis. They often present with non-specific symptoms, if any. Surgical treatment is the best option when a patient’s general condition allows surgery to be performed. It would be interesting to study the genetic and epigenetic profiles of such lesions to see the types and frequencies of the harbored mutations.

## Figures and Tables

**Figure 1 medicina-59-01398-f001:**
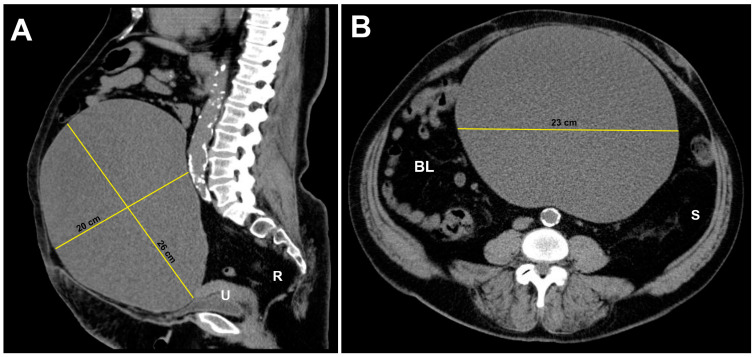
Computed tomography scan of the abdomino-pelvic cavity demonstrating the large extension of the ovarian endometrioma. The intestinal loops were displaced laterally to the right by the cyst. The uterus and urinary bladder were compressed. (**A**) Median sagittal plane. (**B**) Transverse plane. U: uterus; R: rectum; BL: bowel loops; S: sigmoid colon.

**Figure 2 medicina-59-01398-f002:**
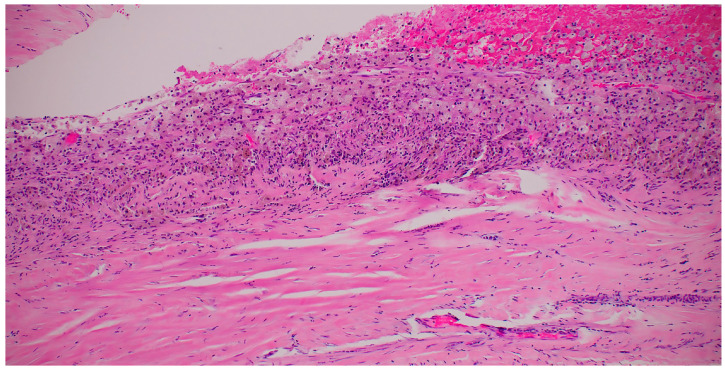
The microscopic examination of the endometrioma’s wall.

**Table 1 medicina-59-01398-t001:** The preoperative laboratory assessment of the patient.

Leukocytes	8.1/nL
Erythrocytes	4.39/pL
Hemoglobin	13.6 g/dL
Hematocrit	41.7%
Mean Corpuscular Volume	95 fL
Mean Corpuscular Hemoglobin	31 pg
Mean Corpuscular Hemoglobin Concentration	32.6 g/dL
Thrombocytes	301/nL
CEA	6.7 ng/mL
CA 125	512.9 U/mL
CA 15-3	17.3 U/mL

## Data Availability

All the data concerning this case report are included in the manuscript.

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
