# Peer review of "A Large Ovarian Endometrioma Occupying the Abdominal Cavity in a Postmenopausal Patient: A Case Report"

_medicina, 2023, doi:10.3390/medicina59081398_

Round 1

Reviewer 1 Report

The authors present a case of Postmenopausal Ovarian Endometrioma in a 60-year old patient with an external diagnosis of abdominal mass. The authors show the computed tomography scan revealing the presence of a cystic mass originating from the right adnexa and measuring 27 cm but fail to depict any other result but rather just mention them.

The abstract and introduction are well written that interests the reader. The case report is in detail but misses on several crucial results which needs to be indicated in the manuscript. 

Figure1: Please highlight any abnormalities by arrow or circle like in the reference 13. Also were the pictures of the gross appearance taken? If so, include in the manuscript similar to reference 13. Include the names os the body parts like in reference 15. Please provide as much as results as possible like the laboratory results showing the ranges of the markers looked at including ovarian tumor markers. Results from the the pathology coming to a conclusion that the tumor was a borderline ovarian tumor. Any comparison images of the left and the right ovary at the time of the surgery. Result from the frozen section biopsy of the ovarian tumor to show no signs of malignancy. Show the result of the microscopic examination of the cyst’s wall confirmed the diagnosis of an ovarian endometrioma with no signs of malignancy. 

The manuscript follows the format of several other published articles on similar topic but does a very poor job in providing good informative results. Since this type of Postmenopausal Ovarian Endometrioma are very rare the authors should take into account to provide as much as results as possible and also do some analysis like they mention about the genetic and epigentic profiling. 

Do the authors know about the patient's previous follow ups if any to determine if the tumor could have been caught earlier. Please mention what precaution a woman should take if detected with Postmenopausal Ovarian Endometriom and when and what tests or examination should be done to prevent the tumor from getting big. 

Please provide the details requested. 

Minor changes to english language on a few occasions. Over good command over the english language.

Author Response

Dear Reviewer,

Thank you so much for your efforts and the time you spent to review our paper. We did our best to adhere to your recommendations and to address your concerns. A point-by-point answer to your comments could be find below:

Comment 1: The authors present a case of Postmenopausal Ovarian Endometrioma in a 60-year old patient with an external diagnosis of abdominal mass. The authors show the computed tomography scan revealing the presence of a cystic mass originating from the right adnexa and measuring 27 cm but fail to depict any other result but rather just mention them. The abstract and introduction are well written that interests the reader. The case report is in detail but misses on several crucial results which needs to be indicated in the manuscript.

Response 1: Thank you very much for your thorough feedback. We truly appreciate the time you spent to evaluate precisely our paper. We will try our best to address all of your concerns.

Comment 2: Figure1: Please highlight any abnormalities by arrow or circle like in the reference 13.

Response 2: Thank you for your instructions. There are no major abnormalities in the abdominal organs similar to those observed in reference 13. The only abnormal findings in our patient is that the bowel loops are displaced. However, we labeled the images with the available findings. Please see the revised figure 1.

Comment 3: Also were the pictures of the gross appearance taken? If so, include in the manuscript similar to reference 13.

Response 3: We fully agree that surgical photos would have been a good addition to this manuscript, but unfortunately, the intraoperative images are unavailable because it is an open surgery. This is mainly due to the internal infection control regulations of our hospital. The pathologist in turn did not take gross images of the specimen before the microscopic examination because it is not a routinely-performed procedure.

Comment 4: Include the names of the body parts like in reference 15.

Response 4: Thank you for your comment. We added the names of the body parts in figure 1 similar to reference 15, as requested.

Comment 5: Please provide as much results as possible like the laboratory results showing the ranges of the markers looked at including ovarian tumor markers.

Response 5: Thank you for your valuable comment. We included the available preoperative laboratory results in an independent table and included the tumor markers in the main text, as requested (lines 69-71 and Table 1).

Comment 6: Results from the pathology coming to a conclusion that the tumor was a borderline ovarian tumor.

Response 6: Thank you for your feedback. Please note the pathologist ruled out the diagnosis of a borderline ovarian tumor and did not come to such a conclusion. The results of the frozen section and the final microscopic examination were ovarian endometrioma. However, we included the microscopic appearance of the endometrioma’s wall (lines 77-80 and Figure 2).

Comment 7: Any comparison images of the left and the right ovary at the time of the surgery.

Response 7: Thank you for your valuable feedback. Unfortunately, intraoperative images are unavailable, as aforementioned.

Comment 8: Result from the frozen section biopsy of the ovarian tumor to show no signs of malignancy.

Response 8: Thank you so much for your comment. The results of the frozen section are already stated in the main text. In addition, the microscopic appearance of the cyst wall was included as previously instructed. Since the results of the frozen section are in accordance with the results of the final pathological examination, we believe it is sufficient to include the picture of the final diagnosis (lines 77-80).

Comment 9: Show the result of the microscopic examination of the cyst’s wall confirmed the diagnosis of an ovarian endometrioma with no signs of malignancy.

Response 9: Thank you so much for your comment. The microscopic appearance of the cyst wall was included as previously instructed (Figure 2).

Comment 10: The manuscript follows the format of several other published articles on similar topic but does a very poor job in providing good informative results. Since this type of Postmenopausal Ovarian Endometrioma are very rare the authors should take into account to provide as much as results as possible and also do some analysis like they mention about the genetic and epigenetic profiling.

Response 10: Thank you so much for your feedback. Please note that we included all the requested information and all the clinically-relevant findings, based on your feedback and the feedback provided by the other two reviewers. However, genetic and epigenetic profiling would have been extremely informative but unfortunately this is beyond our reach due to funding issues and the lack of those investigations in our department.

Comment 11: Do the authors know about the patient's previous follow ups if any to determine if the tumor could have been caught earlier.

Response 11: Thank you for your comment. Unfortunately, the previous examinations and gynecologic visits of the patient are unavailable. However, we expect that regular ultrasonographic scans and early intervention when a continuous growth of the endometrioma would have prevented the endometrioma to reach such a large size. We added this to the discussion section of the manuscript (lines 118-125).

Comment 12: Please mention what precaution a woman should take if detected with Postmenopausal Ovarian Endometrioma and when and what tests or examination should be done to prevent the tumor from getting big.

Response 12: Thank you so much for your valuable comment. We discussed the mentioned points in the discussion section of the manuscript (lines 118-125).

Comment 13: Minor changes to English language on a few occasions. Over good command over the English language.

Response 13: Thank you for your feedback. The written English language was revised carefully as instructed.

Reviewer 2 Report

1. In conclusion, One report documented an ovarian endometrioma of 86 65-cm that weighted 214 Kg. Is 214kg right? Maybe 21.4kg I think.

2. Would you please describe tumor marker including CA-125, CA 19-9?

3. I recommend pathologic pictures improving benign endometrioma.

Author Response

Dear Reviewer,

Thank you so much for your efforts and the time you spent to review our paper. We did our best to adhere to your recommendations and to address your concerns. A point-by-point answer to your comments could be find below:

Comment 1: In conclusion, one report documented an ovarian endometrioma of 86 65-cm that weighted 214 Kg. Is 214kg right? Maybe 21.4kg I think.

Response 1: Thank you so much for your precise feedback. We reviewed the cited reference and according to the author’s statement the weight is 214 KG (471 lbs). Please see the abstract of the first article and the discussion of the second article:

https://pubmed.ncbi.nlm.nih.gov/24953922/

https://www.ncbi.nlm.nih.gov/pmc/articles/PMC9516747/

Comment 2: Would you please describe tumor marker including CA-125, CA 19-9?

Response 2: Thank you so much for your valuable comment. Unfortunately, the CA 19-9 levels were not measured. However, the levels of CA-125, CA 15-3 and CEA were evaluated preoperatively and the results were included in the manuscript (lines 69-71 and table 1).

Comment 3: I recommend pathologic pictures improving benign endometrioma.

Response 3: Thank you for your recommendation. The pathologic picture was added as requested (Figure 2).

Reviewer 3 Report

Title: I suggest changing it because it appears similar to others, also giant is a big word according to the finding size 7630g.

 Page 1, Line 34:  to affect around 10 of reproductive age women…………….. is it percentage? percentage out of the total or what? Worldwide or in a specific country, or age?

 Figure 1 needs labels (the most important ones).

 More results need to be added as the author refers………….. ovarian tumor markers and microscopic examination photos should be added.

 Page 3, Lines 86&87: One report documented an ovarian endometrioma of 65-cm that weighted 214 Kg. This case is thought to be the largest ovarian endometrioma documented in the English literature (15). …………… I didn’t find this in Reference 15 ……… according to the text, it is another reference ………

Shah, A. A., Soomro, N. A., Talib, R. K., Sadhayo, A. N., & Soomro, S. A. (2014). Giant intraabdominal endometrial cyst. Journal of the College of Physicians and Surgeons--Pakistan : JCPSP24(6), 438–440.

 Please check the rest of the references according to the text.

Page 3, Line 100: genetic alterations……………please refer to specific genes that have a role in this.

 Discussion is too simple and needs more explanation.  

Good

Author Response

Dear Reviewer,

Thank you so much for your efforts and the time you spent to review our paper. We did our best to adhere to your recommendations and to address your concerns. A point-by-point answer to your comments could be find below:

Comment 1: Title: I suggest changing it because it appears similar to others, also giant is a big word according to the finding size 7630g.

Response 1: Thank you for your comment. The title was changed as requested.

Comment 2: Page 1, Line 34:  to affect around 10 of reproductive age women……………. is it percentage? percentage out of the total or what? Worldwide or in a specific country, or age?

Response 2: Thank you for your feedback and please excuse our mistake. We intended to write 10% of reproductive aged women. This is the estimated prevalence of endometriosis in reproductive aged women worldwide. We corrected this in the introduction (page 1, line 35).

Comment 3: Figure 1 needs labels (the most important ones).

Response 3: Thank you for your comment. Figure 1 was labeled as requested.

Comment 4: More results need to be added as the author refers………….. ovarian tumor markers and microscopic examination photos should be added.

Response 4: Thank you for your comment. We added the requested information as instructed (lines 69-71, table 1 and figure 2).

Comment 5: Page 3, Lines 86&87: One report documented an ovarian endometrioma of 65-cm that weighted 214 Kg. This case is thought to be the largest ovarian endometrioma documented in the English literature (15). …………… I didn’t find this in Reference 15 ……… according to the text, it is another reference ………

Shah, A. A., Soomro, N. A., Talib, R. K., Sadhayo, A. N., & Soomro, S. A. (2014). Giant intraabdominal endometrial cyst. Journal of the College of Physicians and Surgeons--Pakistan : JCPSP, 24(6), 438–440.

 Please check the rest of the references according to the text.

Response 5: Thank you so much for your precise comment and please excuse this mistake. We admit that we cited the wrong reference accidently. We cited the correct reference (lines 95 and 96).

Comment 6: Page 3, Line 100: genetic alterations……………please refer to specific genes that have a role in this.

Response 6: Thank you for your feedback. We referred to some of the genes that are thought to have a role in the endometriosis growth and behavior, as requested (lines 108 and 109).

Comment 7: Discussion is too simple and needs more explanation.

Response 7: Thank you for your feedback. We extended the discussion based on the recommendations of reviewer 1 (lines 118-125).

Round 2

Reviewer 1 Report

The authors have answered my questions and modified the figures according.